# Ultra-Processed Food Intake and Increased Risk of Obesity: A Narrative Review

**DOI:** 10.3390/foods13162627

**Published:** 2024-08-21

**Authors:** Antonietta Monda, Maria Ida de Stefano, Ines Villano, Salvatore Allocca, Maria Casillo, Antonietta Messina, Vincenzo Monda, Fiorenzo Moscatelli, Anna Dipace, Pierpaolo Limone, Girolamo Di Maio, Marco La Marra, Marilena Di Padova, Sergio Chieffi, Giovanni Messina, Marcellino Monda, Rita Polito

**Affiliations:** 1Department of Human Science and Promotion of Quality of Life, Telematic University San Raffaele, 00166 Rome, Italy; antonietta.monda@unicampania.it; 2Department of Clinical and Experimental Medicine, University of Foggia, 71122 Foggia, Italy; maria.destefano@unifg.it (M.I.d.S.); rita.polito@unifg.it (R.P.); 3Department of Wellness, Nutrition and Sport, Telematic University Pegaso, 80143 Naples, Italy; ines.villano@unipegaso.it (I.V.); fiorenzo.moscatelli@unipegaso.it (F.M.); anna.dipace@unipegaso.it (A.D.); pierpaolo.limone@unipegaso.it (P.L.); 4Department of Experimental Medicine, University of Campania “Luigi Vanvitelli”, 80138 Naples, Italy; salvatore.allocca@unicampania.it (S.A.); maria.casillo@unicampania.it (M.C.); girolamo.dimaio@unicampania.it (G.D.M.); marco.lamarra@unicampania.it (M.L.M.); sergio.chieffi@unicampania.it (S.C.); giovanni.messina@unicampania.it (G.M.); 5Department of Precision Medicine, University of Campania “Luigi Vanvitelli”, 80138 Naples, Italy; antonietta.messina@unicampania.it; 6Department of Economics, Law, Cybersecurity, and Sports Sciences, University of Naples “Parthenope”, 80132 Naples, Italy; vincenzo.monda@uniparthenope.it; 7Department of Humanistic Studies, University of Foggia, 71122 Foggia, Italy; marilena.dipadova@unifg.it

**Keywords:** obesity, ultra-processed foods (UPFs), healthy lifestyle, caloric intake, public health

## Abstract

The prevalence of obesity has become a global health concern, with significant impacts on quality of life and mortality rates. Recent research has highlighted the role of ultra-processed foods (UPFs) in driving the obesity epidemic. UPFs undergo extensive processing, often containing high levels of sugars, fats, and additives, while lacking essential nutrients. Studies have linked UPF consumption to obesity and cardiometabolic diseases, underscoring the importance of dietary patterns rich in whole foods. Thus, the aim of this narrative review is to elucidate the correlation between ultra-processed foods and the increased trend of obesity and its related complications. These foods, prevalent in modern diets, contribute to nutritional deficiencies and excessive caloric intake, exacerbating obesity rates. Lifestyle factors such as busy schedules and quick meal management further drive UPF consumption, disrupting hunger regulation and promoting overeating. UPF consumption correlates with adverse health outcomes, including dyslipidemia, hypertension, and insulin resistance. Promoting whole, minimally processed foods and implementing school-based nutrition education programs are crucial steps. Also, numerous challenges exist, including unequal access to healthy foods, the industry’s influence, and behavioral barriers to dietary change. Future research should explore innovative approaches, such as nutrigenomics and digital health technologies, to personalize interventions and evaluate policy effectiveness. Collaboration across disciplines and sectors will be vital to develop comprehensive solutions and improve public health outcomes globally.

## 1. Introduction

Obesity has been identified as a complex multifactorial disease, the antechamber for the onset of pathological conditions such as diabetes, coronary heart disease, stroke, hypertension, and various tumors affecting the colon, breast, and prostate. Currently, it represents one of the main causes of preventable death: approximately 100,000–400,000 deaths have been estimated every year in the United States, and there has been an increase in health spending amounting to 140–215 billion dollars [1,2,3].The exponential increase in obesity over the last 30 years indicates that weight gain may depend on individual choices in response to the economic environment. The literature has identified several factors that potentially play a crucial role related to the increase in obesity. Prominent among these are higher income levels, the technological evolution of the food industry, the increased prevalence of fast-food restaurants and commercials inducing the population to purchase junk food, possible changes in time preferences, declining smoking rates, an increased prevalence in major depression, and changes in health insurance [4]. To identify dietary factors related to weight gain and obesity, researchers have focused on the choice of dietary patterns [5]. Several epidemiological studies indicate a strong inverse correlation between adherence to a healthy and balanced dietary protocol, such as the Mediterranean model, and an increase in cardiometabolic risk [6,7,8]. Conversely, a diet characterized by highly processed foods is closely associated with obesity and related metabolic comorbidities [9,10]. An understanding of the impact of ultra-processed foods (UPFs) on health and as a risk factor for diet-related diseases, disorders, and conditions is rapidly emerging. In modern society, the role of UPFs proposed by the food industry contrasts with traditional dietary models that promoted the consumption of whole meal products and 0 km foods and emphasized the importance of the locality and seasonality of foods [11,12]. In recent years, the whole food production chain has been significantly influenced by the evolution of technological advances, which have led to the greater accessibility and commercialization of UPFs. The sophistication of production processes has structurally and sensorially modified the nutritional content of foods [13,14] in proportion to the degree of transformation. UPFs are therefore “empty” from a nutritional point of view, but full of added food substances such as sugars, salt, protein isolates, syrups with high fructose content, maltodextrins, and additives such as colorants, flavorings, artificial sweeteners, and thickeners used to enhance the taste of the product, making it palatable, easily usable, and economically accessible. Recently, the view that it is the processing of products that is closely associated with the rise of the obesity epidemic has been gaining more and more credence. This has been verified and supported by a review of prospective cohort studies conducted on more than one million subjects [4]. This led to the development of the NOVA food classification system according to the transformation process, which allows us to identify the dietary drivers related to the increased risk of obesity [15,16,17]. The NOVA classification system provides for a grouping of foods into four categories depending on the degree and extent of processing used [18,19]: unprocessed or minimally processed foods; processed culinary ingredients; processed foods that are produced by adding salt, oil, sugar, or other culinary ingredients to minimally processed foods and processed foods; and finally, at the pinnacle of the processing spectrum, ultra-processed foods (UPFs) [15,16,17,20]. Several studies highlight that UPF consumption is increasing exponentially in recent years and characterizes more than half of the daily calories in US [16], Canadian [19], or British [21] diets. UPF abuse leads to a nutritionally unbalanced diet, rich in calorie-dense foods, saturated fats, sugar, and salt, but low in fiber and micronutrients [21], with a potentially high impact on health [22]. In light of this evidence, the purpose of this narrative review lies in researching the correlation between ultra-processed foods and the increased trend of obesity and related cardiometabolic outcomes.

## 2. Understanding Obesity

### 2.1. Definition and Classification of Obesity

Obesity is a public problem of considerable health importance throughout the world. It affects more than 1 billion people worldwide, of which 650 million are adults [23]. Obesity leads to the onset of metabolic syndrome [24], defined by the National Institutes of Health as a procession of closely related metabolic dysfunctions, including abdominal adiposity, hypercholesterolemia, hypertension, and impaired fasting glycemia [24]. The clinical picture presented is associated with an increased risk of mortality and many non-communicable diseases (NCDs) [24]. The World Health Organization (WHO) defines obesity as “a clinical condition characterized by excessive body weight due to the accumulation of adipose tissue to an extent that negatively affects health” [4,25]. Recent surveys conducted by the WHO on a global scale have shown that obesity is seriously affecting the healthcare system, with risk factors that have led to premature deaths and disabilities (Millennium Burden of Disease Analysis). Therefore, obesity has been recognized as a rapidly expanding global epidemic, and its speed of spread is demonstrated by the prevalence of excess weight (overweight and obesity) doubling between 1980 and 2008 [26]. The late clinical recognition of obesity drastically reduces the possibility of intervention on a clinical, educational, and nutritional level. A system widely used to stratify the population, recognize the degree of severity of the disease, and be easy to use and non-invasive is the Body Mass Index (BMI), an easily calculable indicator defined as the ratio between body weight expressed in kilograms (kg) and height measured in square meters (m^2^). According to the BMI, an underweight individual has a BMI < 18.5, normal weight 18.5 ≤ BMI ≤ 24.99, overweight 25.0 ≤ BMI ≤ 29.99, and obese BMI ≥ 30.0. Obesity is subclassified into Class I obesity (30.0 ≤ BMI ≤ 34.99), Class II (35.0 ≤ BMI ≤ 39.99), and Class III or morbid obesity (BMI ≥ 40.0).

BMI is a less specific measure as it tends to overestimate obesity among subjects with greater muscle mass and underestimate obesity among older adults who have lost muscle mass [27]. To specifically identify obesity, it is necessary to analyze not only the BMI, but also other anthropometric measures: many epidemiological studies have shown that anthropometric measures to identify abdominal obesity such as waist circumference (WC) and waist–hip ratio (WHR) are strong predictors for non-communicable diseases such as type 2 diabetes mellitus [28] and cardiovascular disease (CVD) [29]. World Health Organization (WHO) guidelines state that WC, WHR, and waist-to-height ratio (WHtR) have been shown to be superior predictors to BMI [4,25]. Despite these limitations, BMI continues to be the most commonly used indicator in empirical research on obesity. It is easily measured during a physical examination and cross-referenced with other anthropometric parameters to determine whether the examined subject has a tendency toward obesity. In fact, epidemiological and economic studies on the effects of excess weight, classified according to BMI, indicate that in 2016, obesity levels in the adult population in the United States were estimated at 36.2%, of which 67.9% were overweight, compared to data reported in 1975 where obesity rates did not exceed 11.9%. Even in the pediatric population the trend is rapidly increasing: the prevalence rate of obesity has reached 21.4%. In 2016, Italy recorded an obesity prevalence of 19.4%, with 58.5% of pre-obese individuals (World Health Organization. https://www.who.int/gho/ncd/risk_factors/overweight_obesity/obesity_adults/en/, accessed on 19 August 2024). Costa et al. [30] reported that the prevalence of overweight children in Italy is comparable to that in the United States. It has been estimated that the costs related to excess weight in Italy amount to 4% of the total annual health expenditure of the Italian National Health Service, approximately 4.5 million euros. However, the recorded costs are underestimated due to limitations linked to the classification based on BMI, which does not precisely estimate the prevalence of obesity in the population [31], due to the use of a single scale for both sexes and for all ages; it is therefore an imprecise method, because it may not correspond to the same percentage of body fat [32]. The quantification of adipose tissue with anthropometric measurements, plicometry, and ultrasound is influenced by the distribution of fat by sex and age and by operator-dependent error. Diagnostic imaging, especially computed tomography, nuclear magnetic resonance, biohymoedenziometry, and dual X-ray absorptiometry (DEXA), are considered more accurate and reliable techniques. DEXA is the gold standard for detecting body composition, not only for the degree of accuracy of the results, but also because it is minimally invasive and has an excellent price/detection time ratio. DEXA, a very accurate diagnostic tool, defines the entire body or a segment in terms of the fat, lean, and bone mass [33]. Studies of the Italian population by sex and age indicate a fat mass percentage of 30% for women and 25% for men [34]. In clinical practice, additional techniques such as plethysmography and bio-impedance testing are used, but they cannot be considered diagnostic tools in the same way as DEXA, as they are not very accurate assessments.

Early screening and immediate intervention and prevention policies and strategies have proven to be effective tools to reduce the spread of obesity and related complications, and therefore health and economic spending, with the aim of increasing investments in prevention campaigns, medical training, and the promotion of a healthy and balanced lifestyle [35].

### 2.2. Phenotype of Obesity

There are different obese phenotypes, and this distinction depends on a series of factors: the co-partitioning and diffusion of adipose tissue, genetic predisposition, body composition, the state of inflammation, and the altered metabolism of macronutrients. Therefore, based on the relationship between these factors and their clinical manifestations, four phenotypic typologies have been proposed: normal-weight obese subjects (NWOs), metabolically unhealthy normal-weight obese subjects, metabolically healthy obese subjects, and metabolically unhealthy obese subjects. This new classification is not based on anthropometric measures such as BMI [36]; in fact, the NWO phenotype presents a normal BMI, but a strong reduction in lean mass, low-grade inflammation, a greater cardiovascular risk, hypertension, and metabolic alteration [37]. It might seem like a paradox to define a subject as obese, with a normal weight, but there is an explanation: the distribution of visceral and ectopic fat. In fact, metabolically unhealthy obese subjects have a central fat distribution and are therefore subjected to a greater cardiovascular risk [38]. It is now known that adipose tissue is not a simple, inert storage tissue, but is at the center of a complex molecular and hormonal crosstalk. It is to all intents and purposes an endocrine organ capable of modulating metabolism, hormones, and inflammation through the production of pro- and anti-inflammatory cytokines. Its spread and expansion at the subcutaneous level is followed by visceral and ectopic fat deposition, with a progressive increase in inflammation and infiltration [39]. Obesity therefore involves inflammation at a local level, which then extends systemically. The growth of adipose tissue also influences strength, muscle mass, and bone mass. Therefore, among the previously described phenotypes, for completeness we must also include sarcopenic and osteosarcopenic obesity [40]. 

### 2.3. Factors Contributing to Obesity Epidemic

Numerous studies state that excess weight is a very complex and rapidly expanding public health problem resulting from a combination of individual (genetics, family education) and environmental (unhealthy eating habits, social class) factors [41]. According to the predictions of the World Health Organization for 2030, 30% of deaths in the world will be attributable to the so-called “diseases of abundance” or lifestyle-related diseases, and only early identification and adequate management of the factors of risk triggering the disease, and prevention and disease policies, will be able to reverse this trend. Low rates of physical activity and easy and chronic accessibility to junk food make obesity an “acquired” disease whose harmful causes are to be found in lifestyle and behavioral choices. In addition to biological and behavioral factors, the socioeconomic aspect should not be underestimated: Zhang et al. (2020), in fact, report that in specific geographical areas, the environmental impact, defined as “obesogenic”, strongly increases the probability of developing obesity [42]. 

Additionally, the cultural environment and race likely influence weight. One study highlighted that racial segregation in metropolitan areas was different between the two sexes: there was no obvious association of obesity rates among men, but a positive influence among some women. Specifically, racial segregation for black women was significantly associated with a 1.29 times higher prevalence of obesity (95% CI, 1.00–1.65), while moderately segregated areas had a higher prevalence, 1.35 times higher (95% CI, 1.07–1.70). In contrast, for Mexican-American women living in a highly segregated area, the trend was significantly lower (prevalence ratio, 0.54; 95% CI, 0.33–0.90) [43].

The urban environment also influences obesity rates; in fact, in the first phase of the National Longitudinal Study of Adolescent Health conducted on 20,745 adolescents, it was recorded that the proximity of specific facilities for physical activity in socioeconomically more affluent residential areas showed a correlation with obesity and overweight rates among adolescents who practiced physical activity at least five times a week [44]. Movement is important in weight gain management programs, especially among adolescents, but in school and work environments, the idea that physical activity is not a priority still reigns.

Lack of physical activity, individual behavior, and food choices drastically influence daily calorie intake. Furthermore, in addition to caloric intake, we must also focus on food quality: due to technological progress in food production processes, there has been a strong reduction in the consumption of healthy foods such as whole foods, fruit, vegetables, and fresh products and an increase in the consumption of canned, ultra-processed foods that are unhealthy, as they are rich in hidden ingredients such as fats, simple sugars, salt, and other ingredients that increase their palatability and shelf life, and harmful not only due to the greater caloric intake [45], but above all because they negatively influence the microbial communities that populate our intestine [46].

The microbiome, i.e., the set of our genetic heritage and the environmental interactions of the microorganisms that populate our body (internal and external), influence our metabolism. Emerging evidence demonstrates that the gut microbiota mediates the relationship between host and environment, feeding on the host’s undigested food and in turn producing metabolites and cytokines that influence the host’s metabolism. Nutrition therefore has a crucial impact on the composition of the microbiota, and intestinal microbial imbalance, so-called dysbiosis, is strongly associated with metabolic diseases such as obesity. As mentioned above, the intestinal microbiota influences the host’s metabolism through metabolically active substances produced by it that act on satiety centers, insulin resistance, epigenetic factors, bile acid metabolism, and metabolic signaling (Figure 1). This association proves to be a promising field of study at a therapeutic level to treat, or better yet prevent, obesity [46].

### 2.4. Health Consequences of Obesity: Obesity and Associated Conditions

Obesity is associated with pathological conditions of considerable clinical importance or, worse, pre-existing diseases such as coronary heart disease, neurodegenerative disease, hypertension, stroke, type 2 diabetes mellitus, sleep apnea, and osteoarthritis [47].

#### 2.4.1. Neurodegenerative Disease

As regards neurodegenerative diseases, or pathologies in which the central or peripheral nervous system undergoes progressive degeneration, a strong correlation has been highlighted between obesity and Alzheimer’s disease (AD), Parkinson’s disease (PD), and dementia in the elderly. A recent study [48] found a strong correlation between AD, PD, and metabolic changes induced by diet. Profenno et al. [49] confirm that obesity and diabetes in adults increase the risk of developing AD.

#### 2.4.2. Diabetes

Researchers have estimated that type 2 diabetes mellitus is related to an increased BMI and excess weight in 90% of cases, representing the most associated comorbidity. Obesity and diabetes seem to go hand in hand in metabolically decompensated subjects, and the primum movens is represented by insulin resistance and low-grade inflammation [50,51,52].

#### 2.4.3. Apnea

The respiratory system is greatly affected by excess weight. Respiratory diseases such as asthma and obstructive sleep apnea can arise or, if pre-existing, can worsen with weight gain and appear to be more widespread in overweight/obese children [53].

#### 2.4.4. Autoimmunity

Recent studies record increased rates of diseases involving the immune system in those suffering from obesity. These conditions include rheumatoid arthritis (RA), systemic lupus erythematosus (SLE), inflammatory bowel disease (IBD), multiple sclerosis (MS), type 1 diabetes (T1D), and thyroid autoimmunity (TAI), in particular Hashimoto’s thyroiditis (HT) [54].

#### 2.4.5. Cardiovascular Disease

Excess weight is associated with an increase in cardiovascular diseases (CVDs), but reconnecting with the discussion of the obesity phenotype, normal-weight obese patients may even present a worse prognosis. This phenomenon can represent a paradox. The explanation is to be found in the progressive loss of lean mass, such that obese subjects of normal weight, overweight, and with Class I obesity present a greater cardiovascular risk compared to more serious conditions (Class II, III, and above Class III) [55]. Undoubtedly, overweight and obesity increase the onset of heart failure as they influence the structure and function of the heart, ventricular and diastolic [24,56]. Certainly, excess weight is related to the dysregulation of physiological parameters that lead to the development and worsening of cardiovascular disease, including dyslipidemia, elevated blood sugar, low-grade systemic inflammation, metabolic syndrome, and type 2 diabetes mellitus [24]. One study showed that 23% of heart disease in men and 15% in women was attributable to excess adipose tissue [57] Another study showed that the risk of death increased proportionally with the years of life lived with obesity. Khan et al., in a study with follow up from 1964 to 2015, confirmed that the cardiovascular event was greater in overweight/obese subjects [58].

## 3. Ultra-Processed Food: Definition and Characteristics

The spread of obesity represents a great concern for health, as it is classified as the fifth cause of death worldwide [36]. The latest studies focus on a new perspective: the obesity epidemic seems to derive from a high intake of ultra-processed foods (UPFs). These data emerge from a review of prosthetic cohort studies that analyzed more than 1 million people. UPFs are industrial food formulations that undergo a processing and transformation process through the addition of various ingredients such as simple sugars, fats, salt, and other chemicals such as chemical additives, combined with whole or non-whole foods with the aim of increasing their level of palatability, prolonging their shelf life, and making them convenient and easily available [59,60].

In a recent systematic review, several prospective studies focused on the association between the consumption of ultra-processed foods and the incidence of obesity and cardiometabolic diseases in the adult population [61,62,63,64,65,66]. In most of the articles analyzed, there is an increasing trend of obesity and complications (such as dyslipidemia, hypertension, type 2 diabetes mellitus, and metabolic syndrome) linked to excessive caloric intake, characterized predominantly by packaged and ultra-processed foods to the detriment of a diet rich in whole meal products, fruit, vegetables, and fresh foods.

In 2010, a NOVA classification model was created, which divides foods into four groups based on the degree and type of food processing and transformation [19,67,68]. The first group includes foods that have not undergone a transformation process or have done so only to a minimal extent: for example, processes aimed at ensuring the edibility, suitability for consumption and conservation, safety, and palatability of altered plants and animals (Figure 2).

The second group includes processed ingredients such as butter, oils, salt, and sugar used to make dishes more palatable. The third group includes processed foods, i.e., products to which two/three ingredients from the first and second groups are added to prolong their shelf life and improve their organoleptic quality. Finally, the last classification includes ultra-processed foods (UPFs), i.e., food formulations characterized by the presence of five or more modified or unmodified substances extracted from other foods and combined with minimally whole and non-whole foods, and food additives that improve sensorial quality and prolong shelf life. These include packaged products, breakfast cereals, sweet and savory snacks, packaged bread, margarine, reconstituted meat foods, ready-to-eat soups, ready-to-eat frozen foods, and carbonated and distilled alcoholic beverages. The consumption of UPF foods is spreading exponentially, resulting in nutritional deficiencies (fiber, vitamins, and mineral salts) and a high caloric intake. Several studies show a positive correlation between the consumption of UPF foods and increased rates of obesity and cardiometabolic risk among adult and child populations [61,69,70] (Table 1).

## 4. Link between Ultra-Processed Food Consumption and Obesity

The demand for UPF foods has increased over time, and the causes are to be found in the lifestyle of today’s society: intense work rhythms, the lack of time, and quick meal management lead to a greater demand for foods that can be easily consumed during work activity [8,71,72]. This leads to an unconscious consumption that alters neural and digestive functions, which dysregulate hunger perception and satiety levels, which in turn leads to excessive consumption [73,74]. Furthermore, it has been shown that these foods—precisely because they are poor from a nutritional point of view (in particular in fiber, micronutrients and bioactive compounds [21,63]) but rich in saturated and trans fatty acids, sodium, and refined sugars—reduce satiety levels and therefore increase the glycemic response, altering the insulin response [75], which leads to a greater search for food, favoring a greater energy intake [76] and establishing a serious vicious circle which over time leads to excess weight.

The picture presented determines cardiovascular, metabolic, and blood pressure outcomes that are risky for health. In most UPFs, the salt content is high, and this contributes to increasing the risk of developing hypertension [77]. As previously mentioned, the addition of simple sugars also alters fructose metabolism, in particular favoring insulin resistance first at the liver level and then at the systemic level, exacerbating low-grade inflammation. To overcome insulin resistance and restore glycemic levels, pancreatic β cells produce a greater amount of insulin through a negative feedback mechanism, leading to structural damage and reduced functionality [78]. Furthermore, the excessive consumption of UPFs also leads to a modification of the lipid structure due to the high presence of saturated and trans fatty acids, compromising the catabolism of very-low-density lipoproteins (VLDL-C) and increasing triglycerides in the blood and reducing HDL cholesterol [79]. Finally, the presence of chemical additives, preservatives, and synthetic antioxidants further worsens the clinical picture [80,81]. The packaging used for UPF foods is characterized by releasing chemicals such as bisphenol A, which is harmful to the endocrine system (Figure 2) [82,83,84].

## 5. Ultra-Processed Food and Dietary Patterns

UPF foods differ from non-UPF foods not only in their composition and type of production, but also in their destination. As previously reported, the aim is to obtain artificial, ready-made, quick, easy-to-take, highly palatable foods produced with cheap ingredients [21,67,68]. The production of UPF products contrasts with the recommendations of the dietary model proposed by the Mediterranean diet, which limits the consumption of packaged, processed, and ultra-processed foods. From a nutritional point of view, UPFs appear very unbalanced, composed mostly of ingredients that should be limited, such as added sugars, salt, and saturated and trans fats, and poor in fiber, vitamins, and mineral salts [19]. To paraphrase Paracelsus, it is the dose that makes the poison. In the Mediterranean model, in fact, the consumption of these foods is not demonized, but their intake is limited. Numerous studies conducted on a national scale have recorded a close co-occurrence between the consumption of UPF and the nutritional inadequacy of diets [85]. The appearance of pathological conditions and metabolic complications does not depend only on an excessive daily intake, but on the entirety of the serious nutritional imbalance that derives from the excessive use of these foods. In fact, manufacturing processes generate an inevitable loss of important nutrients for our body and transform others into harmful substances; this happens, for example, with the hydrogenation of polyunsaturated fats. The risk linked to the manipulation and transformation of the food matrix is not limited only to the addition of ingredients such as sugars, vegetable oils, protein isolates, milk powder, additives, emulsifiers etc., but to the formation and release of harmful substances from synthetic packaging [86]. The physico-chemical profile of UPF foods is very complex, so it is reductive to focus on the analysis of a small group of nutrients, which can overlook the evaluation of the substances that are released by the type of ultra-process used. In fact, the analysis often hides the modification at the molecular level to which these products are subjected. Research is moving towards the identification not only of the nutritional profile, but also of composition, chemical structure, and unclear mechanisms that impact health [87]. How does the new formation of toxic substances, chemical hydrogenation, and the presence of synthetic substances influence our biological systems? What are the consequences? At this point, we need to introduce the term “reconstitution”: ingredients used in the manufacturing process are reconstituted through various operations such as the hydrogenation of fats, the extrusion of cereals, and the mechanical extraction of meat which modify the chemical–physical structure of the food matrix, generating new substances [75,76]. The list of ingredients present in UPF foods is very long, among which we also find colorants, artificial sweeteners such as aspartame, saccharin, and acesulfame K, and emulsifiers [88,89]. The use of these substances is not accidental; in fact, they make the product very tasty; they create addiction, leading to greater consumption [65]. They are mistakenly called “foods”, but in reality they are anything but nutritious compared to minimally processed or fresh/whole foods which maintain their own nutritional characteristics. It would be more appropriate to define them as non-nutrient in terms of the substances that are lost and degraded during the processing operations. When the diet is particularly rich in these non-nutrients that replace unprocessed foods, the impact on health is harmful. Ultimately, it is possible to distinguish two dietary models: the first includes minimally processed products, i.e., products or dishes prepared at home or artisanally in restaurants with few processed ingredients or with the addition of processed foods, while the second model includes UPF foods, i.e., drinks and ultra-processed foods such as ready meals and fast food distributed on a large scale [90]. Supermarket shelves are full of this wide range of UPF foods, sweet and savory, and drinks of all kinds, accompanied by claims that attract the attention of the consumer, who is less and less attentive to food quality and consumes his meal quickly during his lunch break. Therefore, the background into which UPFs are inserted contrasts with the model of the Mediterranean diet, which on the contrary encourages consuming meals as a family, around a well-laid table that offers healthy and tasty foods, prepared together. Marketing strategies prove to be very effective in promoting these products and in their combination, such as hamburgers, chips, and carbonated drinks rich in added sugars or snacks and fruit juices, which contain everything other than fruit. The damage that these products generate in the long term on the health of those who feed themselves totally or almost entirely with these products is worrying: a randomized and controlled study recorded that subjects who consumed 80% of their calories from these non-nutrients had a high caloric intake, an increase in adipose tissue, and ultimately severe malnutrition [91] (Table 2) In fact, some studies have revealed a positive correlation between the ultra-processed dietary model and the nutritional scarcity of the diet [92]. Finally, further perspectives are shown in meta-analyses and cohort studies which demonstrate what was previously asserted: dose–response models show in fact that the UPF model can potentially cause damage and become dangerous, leading to the onset of chronic non-communicable diseases and therefore to an increase in mortality when the consumption of these foods becomes dominant.

In conclusion, the aspect that should not be underestimated is the market strategy implemented by companies that produce UPF foods, which use advertising policies aimed at promoting and legitimizing their products, regardless of their nutria-chemical composition and therefore quality. Tighter regulation of these foods is necessary and urgent, in terms of evaluating the nutritional composition and the impact that this dietary model produces on public health. UPF dietary patterns should be reduced, especially in countries that consume them widely and with a view to primary and secondary prevention, especially in the most sensitive age groups, defining safe cut-offs for health [93] (Table 2).

## 6. Public Health Implications

Addressing the global epidemic of obesity requires multifaceted approaches, and public health policies play a pivotal role in shaping individuals’ dietary habits. One critical aspect is curbing the consumption of ultra-processed foods, which are laden with unhealthy additives, sugars, and fats. Governments can implement policies aimed at reducing the availability and accessibility of ultra-processed foods. This includes imposing taxes on sugary beverages and snacks, restricting marketing to children, and regulating portion sizes in food outlets. In addition, establishing clear nutritional guidelines can help consumers make informed choices [48]. Governments can collaborate with health experts to develop guidelines that prioritize whole, minimally processed foods, while discouraging the consumption of ultra-processed options. Educating the public about the health risks associated with ultra-processed foods is essential. Public health campaigns can highlight the adverse effects of excessive sugar, salt, and trans fats on overall health, particularly in relation to obesity, cardiovascular diseases, and diabetes. Also, improving food labeling is crucial for empowering consumers to make healthier choices. Clear and concise labeling that indicates the presence of additives, excessive sugars, and unhealthy fats can help individuals identify ultra-processed products and opt for healthier alternatives. Encouraging the consumption of whole foods, such as fruits, vegetables, whole grains, and lean proteins, is fundamental [46]. Initiatives that make these foods more affordable, accessible, and appealing can shift dietary preferences away from ultra-processed options. It is important, also, to have school-based interventions; indeed, schools play a significant role in shaping children’s dietary habits. Implementing nutrition education programs and ensuring that school meals adhere to health guidelines can promote healthier eating behaviors from a young age [91]. Furthermore, engaging communities through initiatives like farmers’ markets, community gardens, and cooking classes can foster a culture of healthier eating. These interventions not only provide access to fresh, whole foods but also empower individuals with the knowledge and skills to prepare nutritious meals at home. Holding food manufacturers accountable for the nutritional quality of their products is essential. Encouraging reformulation to reduce the levels of added sugars, sodium, and unhealthy fats in processed foods can contribute to improving their overall dietary quality. Combating obesity and reducing the consumption of ultra-processed foods requires a comprehensive approach that encompasses policy changes, education, community involvement, and industry cooperation. By implementing these strategies collectively, societies can promote healthier food environments and mitigate the burden of obesity-related diseases [92].

## 7. Challenges and Future Directions

Addressing the complex issues of obesity and ultra-processed food consumption presents numerous challenges, but also opportunities for innovative research and policy development. One of the greatest challenges is the unequal distribution of resources and access to healthy foods. Low-income communities often have limited access to fresh, nutritious foods and may rely heavily on cheap, ultra-processed options [90]. Addressing these disparities requires multifaceted approaches that consider social, economic, and environmental factors. In addition, the powerful influence of the food industry presents a significant barrier to promoting healthier food environments. Marketing tactics, product placement, and lobbying efforts can perpetuate the consumption of ultra-processed foods, making it challenging to implement effective policies. Future interventions must navigate this landscape, while advocating for public health [91]. Changing dietary behaviors is inherently challenging, requiring sustained motivation, support, and access to resources. Many individuals face barriers such as cultural preferences, lack of cooking skills, and ingrained habits that make it difficult to adopt healthier eating patterns. Future interventions should incorporate behavioral science principles to facilitate lasting change [84]. Ultra-processed foods often have a higher environmental footprint compared to whole, minimally processed alternatives. Addressing obesity while promoting environmental sustainability requires a balance between health outcomes and ecological considerations. Future research should explore strategies for promoting sustainable diets that benefit both human health and the planet. In addition, individual genetic predispositions and biological factors can influence susceptibility to obesity and metabolic disorders [92,93]. Understanding the interplay between genetics, metabolism, and dietary factors is essential for personalized approaches to obesity prevention and treatment. Emerging research areas such as nutrigenomics offer insights into how genetic variation affects responses to dietary interventions. Evaluating the effectiveness of policy interventions is crucial for refining strategies and maximizing their impact. Robust evaluation methods, including longitudinal studies and randomized controlled trials, are needed to assess the outcomes of public health policies targeting obesity and ultra-processed food consumption. Additionally, translating research findings into actionable policies requires collaboration between policymakers, researchers, and stakeholders. Also, the proliferation of digital health technologies offers new avenues for addressing obesity and promoting healthier food choices. Mobile apps, wearable devices, and online platforms can facilitate behavior tracking, provide personalized feedback, and deliver educational content to support individuals in managing their weight and making healthier dietary choices.

In the coming years, research should continue to address these challenges, while exploring innovative approaches to promoting healthier food environments and reducing the burden of obesity-related diseases. Collaboration across disciplines, sectors, and stakeholders will be essential for developing comprehensive solutions that improve public health outcomes on a global scale.

## 8. Conclusions

In conclusion, the link between ultra-processed food consumption and obesity is undeniable, with a mounting body of evidence highlighting the detrimental effects of these foods on public health. The proliferation of ultra-processed foods in modern diets, characterized by their convenience, affordability, and palatability, has contributed to the global obesity epidemic. These foods, laden with unhealthy additives, sugars, and fats, disrupt normal hunger and satiety cues, leading to excessive calorie consumption and metabolic dysregulation. Behavioral science principles should be integrated into interventions to facilitate lasting behavior change, while efforts to promote sustainability must balance health outcomes with environmental considerations. Personalized approaches to obesity prevention and treatment, informed by genetics and metabolism, offer promising avenues for future research. Robust evaluation methods are needed to assess the effectiveness of policy interventions, and digital health technologies can enhance efforts to promote healthier food environments and empower individuals to make informed choices. In the face of these challenges, continued collaboration and innovation are essential for developing comprehensive solutions that improve public health outcomes and reduce the burden of obesity-related diseases on a global scale.

## Figures and Tables

**Figure 1 foods-13-02627-f001:**
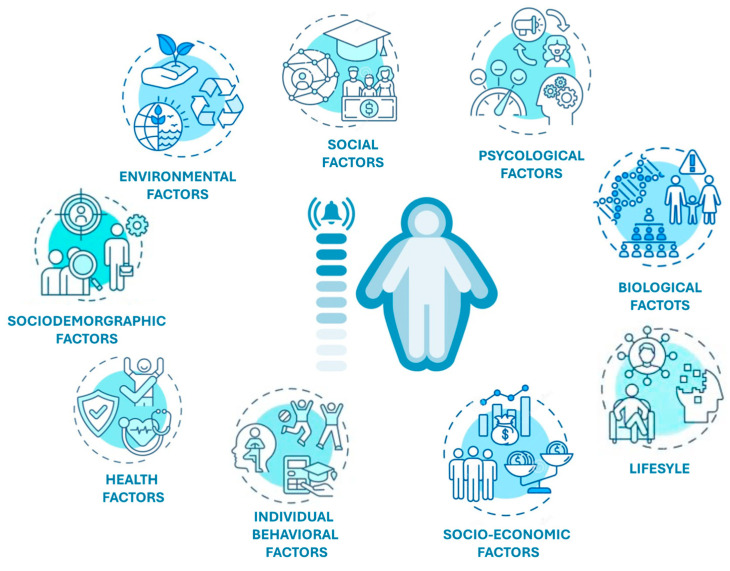
The principal factors leading to obesity.

**Figure 2 foods-13-02627-f002:**
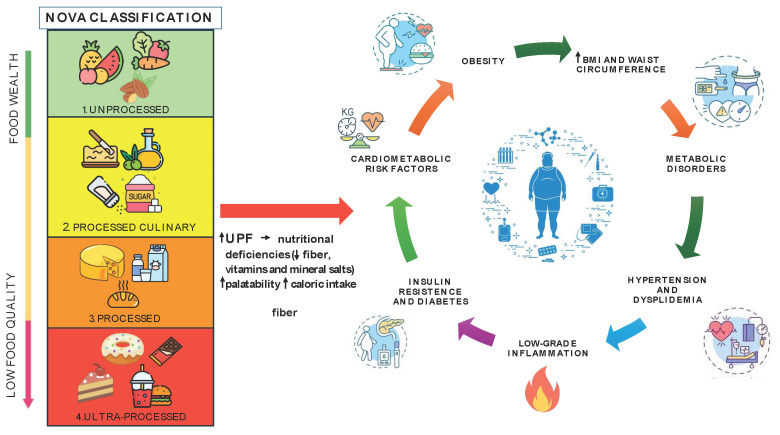
The relationship between UPF foods and obesity.

**Table 1 foods-13-02627-t001:** Key points summarizing the negative effect of UPFs on health and the close correlation between the excessive consumption of UPFs and increased obesity.

Topic	Details
Global Concern	Obesity is the fifth leading cause of death worldwide.
Impact on Quality of Life	Excess weight compromises quality of life and affects over 1 billion people globally, including 650 million adults.
Obesity and UPFs	Recent studies suggest the obesity epidemic may be driven by the high intake of ultra-processed foods (UPFs). UPFs are rich in calories but low in nutrients, contributing to metabolic disorders.
Nature of UPFs	Industrial food formulations processed with added sugars, fats, salt, and chemicals to increase palatability, shelf life, and convenience.
NOVA Classification Model:	Created in 2010, it divides foods into four groups based on processing degree and type. Unprocessed or minimally processed foods for edibility, consumption suitability, conservation, safety, and palatability (Group 1). Processed ingredients (e.g., butter, oils, salt, sugar) used to enhance palatability (Group 2). Processed foods with added ingredients from Groups 1 and 2 to prolong shelf life and improve organoleptic quality (Group 3). Ultra-processed foods (UPFs) with five or more ingredients, including additives for sensory quality and shelf life (Group 4).
Examples of UPFs	Packaged products, breakfast cereals, snacks, packaged bread, margarine, reconstituted meat foods, ready-to-eat soups and frozen foods, carbonated and distilled alcoholic beverages.
Nutritional Impact of UPFs	Lead to nutritional deficiencies (fiber, vitamins, minerals) and high caloric intake.
Health Correlation	Positive correlation between UPF consumption and higher rates of obesity and cardiometabolic risk in both adults and children.

**Table 2 foods-13-02627-t002:** Main characteristics which summarize the chemical and nutritional composition of UPFs.

Topic	Details
UPF Production and Composition	UPFs are made with cheap ingredients, aiming to be ready-made, quick, easy-to-take, and highly palatable.
Nutritional Imbalance of UPFs	High in added sugars, salt, saturated, and trans fats; low in fiber, vitamins, and minerals.
Mediterranean Diet Contrast	Limits consumption of packaged, processed, and ultra-processed foods. Promotes fresh, minimally processed foods.
Nutritional Inadequacy	UPFs linked to unbalanced diets and pathological conditions due to excessive intake and nutritional imbalance.
Nutrient Loss and Harmful Substances	Manufacturing processes of UPFs cause nutrient loss and creation of harmful substances (e.g., hydrogenation of fats).
Risks from Packaging	Harmful substances can be released from synthetic packaging used for UPFs.
Complexity of UPFs	Physico-chemical profile is complex, often hiding harmful modifications at the molecular level.
Reconstitution in UPFs	Ingredients are reconstituted through processes like hydrogenation, extrusion, and mechanical extraction, altering the food matrix.
Additives in UPFs	Includes colorants, artificial sweeteners (aspartame, saccharin, acesulfame K), and emulsifiers. These enhance taste and create addiction.
Health Impact	UPF-rich diets lead to high caloric intake, increased adipose tissue, severe malnutrition, and chronic non-communicable diseases.
Regulation Needs	Urgent need for stricter regulation on UPFs, evaluating their nutritional composition and public health impact.
Prevention Strategies	Reduce UPF consumption, especially in countries with high intake. Define safe cut-offs for sensitive age groups.

## Data Availability

No new data were created or analyzed in this study. Data sharing is not applicable to this article.

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
