# Peer review of "Ultra-Processed Food Intake and Increased Risk of Obesity: A Narrative Review"

_foods, 2024, doi:10.3390/foods13162627_

Round 1
Reviewer 1 Report
Comments and Suggestions for Authors
The article entitled "Ultra-Processed Food Intake and Increased Risk of Obesity: A Narrative Review" explores the correlation between ultra-processed foods (UPFs) and the rising prevalence of obesity. It provides an extensive literature review covering the definitions, characteristics, and health impacts of UPFs, and discusses public health implications and future research directions. However, it has several notable deficiencies in its structure, content, and methodology descriptions. Therefore, I recommend rejecting the manuscript in its current form.
Major comments:
1. Reduce the extensive detail on the definition and characteristics of obesity and provide more balanced coverage on the description and impact of UPFs.
2. Shorten the introduction and conclusion to eliminate redundancy and focus on key points.
3. Provide brief evaluations of the methodologies of the key studies cited to help readers assess the quality and reliability of the research findings.
4. Replace outdated references with more recent studies to ensure the article reflects the latest research and trends in the field.
5. Add charts and graphs to visually present data and conclusions, such as the correlation between UPF consumption and obesity rates.
Author Response
Dear reviewer,
thank you for your precious suggestions and time.
Please, in attached file our response.
Best regards.

Reviewer 2 Report
Comments and Suggestions for Authors
Nicely done commentary and review of the connection between obesity increases and ultra processed foods. Generally well-referenced although there are some areas that could use additional references. For example, in the first paragraph of the Introduction, you don't provide references for your statement about the literature noting factors associated with rising obesity.
Also, there is quite a bit of redundancy. For example, in Part 3, where you begin to discuss UPF, you again mention the obesity epidemic and the sequelae which you have done previously. The paper could benefit from some editing down to make it more readable.
Comments on the Quality of English Language
There are some sentences which need some minor edits as the English should be improved. Here is an example in the abstract: "It is fundamental promoting whole, minimally processed food...."
better to say "It is fundamental to promote whole, minimally processed food...."
Author Response

(The authors gave the same response as above.)

Reviewer 3 Report
Comments and Suggestions for Authors
The review by Monda and de Stefano et al. outlines a very important issue that we face in our modern and technologically advanced society. Overall, despite some minor writing and formatting problems, the review is well written. I have one major point that I would like to clarify below.
Highly recommended
The review would benefit from one or two concrete examples of how, for example, the introduction of a sugar tax has had a positive impact on nutrition. Studies on this should already be available for the UK. An outline of existing pilot projects to improve nutrition would add some substance to the review.
Minor Revision
Page 2, line 92 to page 3 line 108: The information given here is duplicated, the readability is not good and the four categories are not clear to understand. Please rephrase.
Page 4, line 201 and line 201: Please start the sentences with capital letters.
Figure 1: Please correct „FACTOTS“ to „FACTORS“.
Page 6, line 272: Please complete the sentence „cause of the factor linked to type 2 diabetes mellitus [50].“
Page 7, line 305: „al“ is an abbreviation, please write „al.“ with a full stop.
Figure 2: Please correct „WEIST“ to „WAIST“.
Page 11, line 443: Please start the sentences with capital letters.